Evaluation of five regions as DNA barcodes for identification of Lepista species (Tricholomataceae, Basidiomycota) from China

Wang Siyu 1
Guo Hongbo 2
Li JiaJia 1
Li Wei 1
Wang Qin 3
Yu Xiaodan 1 yuxd126@126.com
1 College of Biological Science and Technology, Shenyang Agricultural University , Shenyang , China
2 College of Life Engineering, Shenyang Institute of Technology , Fushun , China
3 Liaoning Academy of Forestry , Shenyang , China
Keller Nancy
Electronic publication date: 2019 Jul 15
Publication date: 2019
Volume: 7
Electronic Location ID: e7307
Received 2019 Mar 19; Accepted 2019 Jun 17
Copyright: © 2019 Wang et al.
Copyright year: 2019
Copyright holder: Wang et al.
License: This is an open access article distributed under the terms of the Creative Commons Attribution License, which permits unrestricted use, distribution, reproduction and adaptation in any medium and for any purpose provided that it is properly attributed. For attribution, the original author(s), title, publication source (PeerJ) and either DOI or URL of the article must be cited.
License URL: https://creativecommons.org/licenses/by/4.0/

Keywords: Agaricales, Intra-specific diversity, DNA barcoding, Inter-specific diversity, Species delineation

Funding: National Natural Science Foundation of China 31200011, 31770014 Natural Science Foundation of Liaoning Province Science and Technology Department 201602668 This study was supported by the National Natural Science Foundation of China (No. 31200011, 31770014) and the Natural Science Foundation of Liaoning Province Science and Technology Department (201602668). The funders had no role in study design, data collection and analysis, decision to publish, or preparation of the manuscript.

==============================
Background

Distinguishing among species in the genus Lepista is difficult because of their similar morphologies.

Methods

To identify a suitable DNA barcode for identification of Lepista species, we assessed the following five regions: internal transcribed spacer (ITS), the intergenic spacer (IGS), nuclear ribosomal RNA subunit, mitochondrial small subunit rDNA, and tef1. A total of 134 sequences from 34 samples belong to eight Lepista species were analyzed. The utility of each region as a DNA barcode was assessed based on the success rates of its PCR amplification and sequencing, and on its intra- and inter-specific variations.

Results

The results indicated that the ITS region could distinguish all species tested. We therefore propose that the ITS region can be used as a DNA barcode for the genus Lepista. In addition, a phylogenetic tree based on the ITS region showed that the tested eight Lepista species, including two unrecognized species, formed eight separate and well-supported clades.

Introduction

Lepista (Fr.) W.G. Sm., a genus in the family Tricholomataeae, was erected by Smith in 1870 and contains about 50 species (Kirk et al., 2008). A total of 12 Lepista species have been reported in China where they are widely distributed (Mao, 2000; Li et al., 2011). Some Lepista species are popular edible mushrooms in China, and these include Lepista nuda (Bull.) Cooke, L. sordida (Schumach.) Singer, and L. irina (Fr.) H.E. Bigelow (Dai et al., 2010).

The genus Lepista can be distinguished from other genera by the coarse surface of its spores, a white to pale-pink spore print, and clamped hyphae (Singer, 1986; Bon, 1987). Within the genus, however, the limited morphological characteristics make it difficult to distinguish among the species. As a result, misidentification is common both between and within species of Lepista. For example, L. irina and L. panaeola have a similar whitish pileus. According to Alvarado et al. (2015), the two species differ in spore size and spore wall structure but the assessment of these characters varies among observers. In addition, other morphological characteristics including the color of the pileus often vary with environmental factors. The pileus of L. nuda, for example, was described as gray brown or russet brown in some studies but as purple brown in others (Bon, 1987; Hansen & Knudsen, 1992).

Accurate identification of species is important for conserving the genetic resources of Lepista, and rapid and reliable species identification is now possible via DNA barcoding. DNA barcoding, which uses short DNA sequences of standard genomic regions, has become increasingly important in identifying species (Badotti et al., 2017; Li et al., 2017) and discovering new species (Zhao et al., 2011; Al-Hatmi et al., 2016). DNA barcoding also could provide the primary information for species delimitation in poorly known groups (Vogler & Monaghan, 2007) and help identify candidate exemplar taxa for a comprehensive phylogenetic study (Hajibabaei et al., 2007). Based on the requirements for standardized DNA barcoding, the sequences of all candidate markers should be short and should have high rates of successful amplification and high rates of successful sequencing. DNA barcoding also requires that candidate markers have substantial inter-specific variation but not intra-specific variation. The internal transcribed spacer (ITS) region of the nuclear ribosomal RNA gene has been used as a general barcode marker for some groups in the Basidiomycota (Dentinger, Didukh & Moncalvo, 2011; Cai, Tang & Yang, 2012; Buyck et al., 2014; Badotti et al., 2017). Other candidate segments that have been used as barcoding markers for mushrooms previously, including the mitochondrial cytochrome oxidase I gene (cox1) (Vialle et al., 2009), the second subunit of RNA polymerase II (RPB2) (Li et al., 2017), and the β-tubulin and elongation factor 1-α (tef1) (Guo et al., 2016).

The goal of the present study was to test the utility of DNA barcodes to the identification of the Lepista species as edible species to address the question, that is, due to the limited morphological characteristics within the genus Lepista, misidentification often happened. To address the question, we evaluated the following five markers as DNA barcodes for identification of eight Lepista species: ITS region, the intergenic spacer (IGS), the large nuclear ribosomal RNA subunit (nLSU), the mitochondrial small subunit rDNA (mtSSU), and tef1.

Materials and Methods

Ethics statement

Lepista species are neither protected nor endangered in the sampled areas, and all samples were collected by researchers following current Chinese regulations. None of the sampled locations are privately owned or protected by law.

Sampling

In a previous study (Alvarado et al., 2015), the genus Lepista was divided into three clades. The current study included two species from each of the three clades plus two unidentified Lepista species. A total of 34 samples of the eight Lepista species were collected from September 2012 to August 2017 (Table 1). Tissue blocks were removed from the inner part of the fresh basidiomata for DNA analyses. The specimens were dried with an electric air ventilation drier and deposited in the Fungal Herbarium of Shenyang Agricultural University (SYAU-FUNGI).

Table 1 The Lepista samples used in this study.

Taxon	Specimen vouchera	ITSb	IGSb	nLSUb	mtSSUb	tef1b	
Lepista densifolia	SYAU-FUNGI-022	MK116588	MK389519	–	MK389570	–	
Lepista irina	SYAU-FUNGI-023	MK116589	MK389520	MK389546	MK389571	MK551215	
Lepista irina	SYAU-FUNGI-024	MK116590	MK389521	MK389547	MK389572	MK551216	
Lepista irina	SYAU-FUNGI-025	MK116591	–	MK389548	MK389573	–	
Lepista nuda	SYAU-FUNGI-021	MH428843	MK389523	MK389549	MK389575	MK440311	
Lepista nuda	SYAU-FUNGI-026	MK116594	–	–	–	MK440315	
Lepista nuda	SYAU-FUNGI-017	MH428839	MK389524	MK389550	MK389576	MK440312	
Lepista nuda	SYAU-FUNGI-019	MH428841	–	MK389551	MK389577	MK440313	
Lepista nuda	SYAU-FUNGI-027	MK116593	MK389525	MK389552	MK389578	MK440314	
Lepista nuda	SYAU-FUNGI-014	MH428836	MK389526	MK389553	MK389579	MK440315	
Lepista nuda	SYAU-FUNGI-028	MK116595	–	MK389554	MK389580	MK440317	
Lepista nuda	SYAU-FUNGI-029	MK116592	MK389522	–	MK389574	MK440310	
Lepista panaeola	SYAU-FUNGI-030	MK116597	MK389527	–	MK389581	–	
Lepista panaeola	SYAU-FUNGI-031	MK116598	–	–	–	–	
Lepista panaeola	SYAU-FUNGI-032	MK116599	MK389529	–	MK389583	MK551218	
Lepista panaeola	SYAU-FUNGI-033	MK116600	MK389530	MK389555	MK389584	–	
Lepista panaeola	SYAU-FUNGI-034	MK116601	MK389531	MK389556	MK389585	–	
Lepista panaeola	SYAU-FUNGI-035	MK116596	MK389528	MK389557	MK389582	MK551217	
Lepista saeva	SYAU-FUNGI-036	MK116602	MK389532	MK389558	MK389586	–	
Lepista saeva	SYAU-FUNGI-037	MK116603	MK389533	MK389559	MK389587	–	
Lepista saeva	SYAU-FUNGI-038	MK116604	MK389534	MK389560	MK389588	–	
Lepista sordida	SYAU-FUNGI-039	MK116605	MK389535	MK389561	MK389589	MK551219	
Lepista sordida	SYAU-FUNGI-040	MK116606	MK389536	–	MK389590	–	
Lepista sordida	SYAU-FUNGI-041	MK116607	MK389537	MK389563	MK389591	–	
Lepista sordida	SYAU-FUNGI-042	MK116609	MK389539	MK389564	MK389594	MK551221	
Lepista sordida	SYAU-FUNGI-043	MK116610	MK389540	MK389565	MK389593	MK551222	
Lepista sordida	SYAU-FUNGI-044	MK116608	MK389538	MK389562	MK389592	MK551220	
Lepista sp 1	SYAU-FUNGI-045	MK116611	–	–	–	MK440305	
Lepista sp 1	SYAU-FUNGI-046	MK116612	–	MK389567	–	MK440306	
Lepista sp 1	SYAU-FUNGI-047	MK116613	MK389541	MK389568	MK389597	MK440307	
Lepista sp 1	SYAU-FUNGI-048	MK116614	MK389542	MK389566	MK389595	MK440308	
Lepista sp 1	SYAU-FUNGI-049	MK116615	MK389543	–	MK389596	MK440309	
Lepista sp 2	SYAU-FUNGI-050	MK116617	MK389544	–	–	–	
Lepista sp 2	SYAU-FUNGI-051	MK116616	MK389545	MK389569	–	–	
Notes:

a SYAU-FUNGI: Fungal Herbarium of Shenyang Agricultural University, Shenyang, China;

b GenBank accession numbers in bold indicate the sequences generated in this study.

Morphological observations

Morphological identification was based on previous studies (Singer, 1986; Bon, 1987; Li et al., 2015). Microscopic characteristics of the basidiomata were assessed by examining dried specimens that had been treated with 5% KOH solution and Melzer’s reagent with a light microscope.

DNA extraction, amplification, and sequencing

Genomic DNA was extracted from fresh blocks of tissue with a plant DNA extraction kit (Sunbiotech, Beijing, China). Crude DNA extracts were used as templates for PCR, and a total of five primers were used for amplification (Table 2). Reaction mixtures were as described by Yu et al. (2014). For the amplification of ITS, IGS, nLSU, and mtSSU, the PCR conditions consisted of an initial denaturation at 94 °C for 2 min; followed by 30 cycles of denaturation at 94 °C for 35 s, annealing at 45 °C for 35 s, and extension at 72 °C for 1 min; and a final extension at 72 °C for 10 min. For tef1, the PCR protocol consisted of initial denaturation at 94 °C for 2 min; followed by 10 cycles at 94 °C for 35 s, 57 °C for 35 s (decreasing 0.3 °C per cycle), and 72 °C for 1 min; followed by 29 cycles at 94 °C for 35 s, 54 °C for 35 s, and 72 °C for 1 min; and a final extension at 72 °C for 10 min. PCR products were checked on a 1.0% agarose gel and visualized by staining with ethidium bromide. Sequencing was performed on an ABI Prism 3730 genetic analyzer (PE Applied Biosystems, Foster City, CA, USA). The sequences generated from this study are listed in Table 1.

Table 2 Primers used in this study.

Regions	Primer	Sequence (5′-3′)	Reference	
ITS	ITS5	GGA AGT AAA AGT CGT AAC AAG G	White et al. (1990)	
ITS4	TCC TCC GCT TAT TGA TAT GC	White et al. (1990)	
IGS	CNL12	CTG AAC GCC TCT AAG TCA G	White et al. (1990)	
	5SA	CAG AGT CCT ATG GCC GTG AT	White et al. (1990)	
nLSU	LROR	ACC CGC TGA ACT TAA GC	Rehner & Samuels (1994)	
LR7	TAC TAC CAC CAA GAT CT	Vilgalys & Hester (1990)	
mtSSU	MS1	CAG CAG TCA AGA ATA TTA GTC AAT G	White et al. (1990)	
MS2	GCG GAT TAT CGA ATT AAA TAA C	White et al. (1990)	
tef1	tefF	TAC AAR TGY GGT GGT ATY GAC A	Morehouse et al. (2003)	
tefR	ACN GAC TTG ACY TCA GTR GT	Morehouse et al. (2003)	

Data analyses

Sequences of each region were aligned with Clustal X (Thompson et al., 1997) and then manually edited with BioEdit 5.0.6 (Hall et al., 2003). The aligned sequences of each region were analyzed using DNAstar 7.1.0 (Lasergene, WI, USA) to calculate the similarity matrices. The intra- and inter-specific variations of the candidate barcode loci for each species were then assessed using TaxonGap 2.4.1 (Slabbinck et al., 2008). Finally, the results were processed and showed by GSview 4.9.

Genetic pairwise distances for evaluating the sequence variations within and between species of the potential barcode regions were computed using MEGA 7.0 (Kumar, Stecher & Tamura, 2016) based on the Kimura 2-Parameter (K2P) model (Kimura, 1980). Barcoding gaps comparing the distributions of the pairwise intra- and inter-specific distances for each candidate barcode with distance intervals of 0.004 (ITS, nLSU, and mtSSU) or 0.008 (IGS and tef1) were estimated in Microsoft Excel 2016.

Neighbor-joining tree reconstruction

To show the relationships among the eight Lepista species, a neighbor-joining tree was constructed based on the ITS region using MEGA with the K2P substitution model. Branch support was calculated by a bootstrap analysis with 1,000 replicates, and Tricholoma matsutake (AB699640) was used as the outgroup. For comparison, the combined dataset of five regions was used to construct a neighbor-joining tree. Alignments have been deposited in TreeBASE (http://purl.org/phylo/treebase/phylows/study/TB2:S24378).

Results

PCR amplification and sequencing

A total of 134 sequences of the five candidate DNA barcode regions were obtained from the eight Lepista species (Table 1). The five regions were then evaluated for their potential as barcoding markers (Table 3). Sequence lengths ranged from 400 bp for IGS to 1,000 bp for nLSU, that is, all five regions were sufficiently short to be used as barcode markers. The amplification success rate exceeded 90% for all regions except tef1, and the sequencing success rate was highest (100%) for ITS.

Table 3 Results of the amplification and sequencing of five regions in the genomes of eight Lepista species.

Region	Region length (bp)	Total number of samples	No. of PCR successes	PCR success rate (%)	No. of sequencing successes	Sequencing success rate (%)	
ITS	605–615	34	34	100	34	100	
IGS	415–440	34	34	100	27	79	
nLSU	934–939	34	33	97	24	71	
mtSSU	662–740	34	32	94	28	82	
tef1	861–920	34	23	68	21	62	

Intra- and inter-specific variation

According to TaxonGap analyses of the intra- and inter-specific variations of the candidate DNA barcode regions, ITS, IGS, tef1, and mtSSU provided a somewhat better resolution of the eight species than nLSU. Except for nLSU, the other four regions showed significant inter- and intra-specific variation (Fig. 1).

Figure 1 Intra- and inter-specific variations among the candidate barcode regions (ITS, IGS, nLSU, mtSSU, and tef1) from eight Lepista species.

Graphs were generated by TaxonGap software. The black and gray bars represent the level of inter- and intra-specific variations, respectively. The thin black lines indicate the lowest inter-specific variation for each candidate barcode. Taxon names next to the dark bars indicate the most closely related species among the species listed on the left. Four regions, that is, ITS, IGS, tef1, and mtSSU, showed significant inter- and intra-specific variation.

Barcoding gaps

Three regions, that is, ITS (Fig. 2A), IGS (Fig. 2B), and tef1 (Fig. 2E), had relatively clear barcoding gaps. The two remaining candidate barcodes (mtSSU and nLSU) had overlaps between their intra- and inter-specific distances (Figs. 2C and 2D).

Figure 2 Frequency distributions of intra- and inter-specific Kimura-2-Parameter pairwise distances among ITS, IGS, nLSU, mtSSU, and tef1 datasets from eight Lepista spp.

The black and gray bars represent the level of intra- and inter-specific variations, respectively. Three regions, that is, ITS, IGS, and tef1, had relatively clear barcoding gaps. (A) ITS. (B) IGS. (C) nLSU. (D) mtSSU. (E) tef1.

Neighbor-joining analysis

In a tree generated by a neighbor-joining analysis of the ITS region, the eight species were well-separated from each other and formed independent terminal branches (Fig. 3). Sequences from different samples of the same species showed high bootstrap values. Two clades, named Lepista sp 1 and L. sp 2, were supported by high bootstrap values and were inferred to represent new species of Lepista. The topology of the combined dataset tree was similar to that produced by ITS region (Fig. S1).

Figure 3 A neighbor-joining tree generated by analysis of ITS from eight Lepista spp.

Bootstrap values ≥70% are shown above the relevant branches. The eight Lepista spp. are highlighted in bold.

Discussion

There are two important factors for evaluating candidate DNA barcodes: a high success rate of PCR amplification and sequencing, and substantially greater inter-specific than intra-specific variation. In the current study, the ITS region had high success rates of amplification and sequencing, substantially greater inter-specific than intra-specific variation, as well as clear barcoding gaps among the Lepista species. Based on the criteria, we therefore conclude that the ITS region would be useful for the identification of Lepista species and determine that the ITS is a suitable DNA barcode for the genus Lepista.

The ITS region has been proposed as a universal barcode for fungi (Schoch et al., 2012). The region is present in several chromosomes and is arranged in tandem repeats that are thousands of copies long (Ajmal Ali et al., 2014). Because of the high copy number, the ITS region is easy to amplify and sequence, even with samples from very old specimens (Larsson & Jacobsson, 2004). ITS has been found to be a suitable barcode for some groups in the Agaricales, including the genus Cortinarius (Liimatainen et al., 2014; Stefani, Jones & May, 2014) and the family Lyophyllaceae (Bellanger et al., 2015).

Although IGS had a high PCR success rate (100%) and suitable inter- and intra-specific variation, its sequencing success rate was relatively low (82%), which made it the second best marker after ITS for identification of Lepista species. IGS has been previously used to differentiate among species and even among strains within the same species in yeasts (Fell et al., 2000; Scorzetti et al., 2002). In the current study, the regions of nLSU and mtSSU lacked barcoding gaps in the analysis of intra- and inter-specific distance. tef1 showed clear barcoding gaps, but its amplification and sequencing success rates were low.

In preliminary studies, we also assessed the largest subunit of RNA polymerase II (RPB1) and RPB2, but we obtained only six sequences of RPB2 and one sequence of RPB1. These numbers of RPB1 and RPB2 sequences were too small for analysis of barcoding, and the two regions were therefore not included in this study.

According to the phylogenetic analysis based on the ITS region, the eight Lepista species received high support (≥98%), which demonstrates that ITS could be used for the identification of Lepista species. The two new clades identified in the present study may represent two new species. Identification of cryptic species by DNA barcoding has been reported in the other groups, such as Amillariella (Guo et al., 2016) and Pleurotus (Li et al., 2017). In future research, the morphological characteristics of Lepista sp 1 and L. sp 2 should be described, and the utility of ITS as a barcode for identification of additional Lepista species should be evaluated.

Conclusions

In this study, we assessed five regions for identifying a DNA barcode for eight Lepista species. Only the ITS region had the highest success rates of amplification and sequencing, substantially greater inter-specific than intra-specific variation. Therefore, we propose that the ITS region could be used as a suitable DNA barcode for the genus Lepista. And the ITS region also could separate all the tested Lepista species in the phylogenetic analyses. Overall, the ITS region was proved as a reference marker for the other species.

Supplemental Information

Supplemental Information 1 A neighbor-joining tree generated by analysis of five regions from eight Lepista spp.

Figure S1. A neighbor-joining tree generated by analysis of five regions from eight Lepista spp. Bootstrap values ≥70% are shown above the relevant branches. The eight Lepista spp. are highlighted in bold.

Click here for additional data file.

Supplemental Information 2 The 30 ITS sequences used in this study.

Click here for additional data file.

Supplemental Information 3 The 27 IGS sequences used in this study.

Click here for additional data file.

Supplemental Information 4 The 24 nLSU sequences used in this study.

Click here for additional data file.

Supplemental Information 5 The 28 mtSSU sequences used in this study.

Click here for additional data file.

Supplemental Information 6 The 21 tef1 sequences used in this study.

Click here for additional data file.

Supplemental Information 7 ITS tree.

Click here for additional data file.

Supplemental Information 8 Combined regions tree.

Click here for additional data file.

We thank Prof. Bruce Jaffee for correcting the English.

Additional Information and Declarations

Competing Interests

Author Contributions

Data Availability

The authors declare that they have no competing interests.

Siyu Wang performed the experiments, approved the final draft.

Hongbo Guo analyzed the data, approved the final draft.

JiaJia Li contributed reagents/materials/analysis tools, approved the final draft.

Wei Li contributed reagents/materials/analysis tools, approved the final draft.

Qin Wang prepared figures and/or tables, approved the final draft.

Xiaodan Yu conceived and designed the experiments, authored or reviewed drafts of the paper, approved the final draft.

The following information was supplied regarding data availability:

Raw data are available at NCBI Genbank via accession numbers MK116588– MK116617, MK389519–MK389597, MK440305–MK440317, and MK551215–MK551222. Alignments are available in TreeBASE (http://purl.org/phylo/treebase/phylows/study/TB2:S24378).

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
