# Peer review of "Evaluation of five regions as DNA barcodes for identification of Lepista species (Tricholomataceae, Basidiomycota) from China"

_PeerJ, doi:10.7717/peerj.7307_

## Round 0.1 · original submission · Minor Revisions

As you can see, your article has been reviewed by three experts. In general they are positive on the manuscript but suggest some editing changes, particularly as suggested by reviewers 1 and 3.

Reviewer 1 ·

Basic reporting

see below

Experimental design

see below

Validity of the findings

see below

Additional comments

The use of ITS as a barcoding marker for a genus of mushroom forming fungus, Lepista is presented in this manuscript. Overall, this is a good paper and contributes to numerous other clades of fungi where the ITS region is a favorable barcoding marker.

Some comments to help the authors improve the quality of their paper are listed below:


1. Line 47, remove "(" before However. Line 48, delete 'believed', and then I think the sentence will make better sense.

2. Line 47–48, "two physical" rewrite as " as the two morphological characteristics...the sentence is unclear, please rewrite

3. Line 49, delete "some"

4. Line 54–57, DNA barcoding is different from phylogenetic analysis - please reframe your sentences here...

See these papers: Hajibabaei, M., Singer, G.A., Hebert, P.D. and Hickey, D.A., 2007. DNA barcoding: how it complements taxonomy, molecular phylogenetics and population genetics. TRENDS in Genetics, 23(4), pp.167-172
Vogler, A.P. and Monaghan, M.T., 2007. Recent advances in DNA taxonomy. Journal of Zoological Systematics and Evolutionary Research, 45(1), pp.1-10.

5. Lines 61–63, please cite: Dentinger, B.T., Didukh, M.Y. and Moncalvo, J.M., 2011. Comparing COI and ITS as DNA barcode markers for mushrooms and allies (Agaricomycotina). PLoS One, 6(9), p.e25081. This is a key paper.

6. cox 1 has found to be unsuitable for fungi, so please don't refer in that it was judged to be a suitable barcoding marker...rather say it has been used for mushrooms previously...

7. Line 67, ITS, Internal transcribed spacer region.

8. Fig.1 Legend, explain the key results rather than simply stating what it means...Writing the legend with the key results will make manuscript more interesting for the reader. Same for Fig. 2.

9. Line 184, Lepista should be italics...

10. Are there type species in the samples analyzed? If so, please indicate. Also, how many species of Lepista have been sequenced? The work is based only on <10 species out of possibly 50 - does that give a good indication for the group? Have there been other studies of DNA barcoding on this group?

11. In Fig. 3, please only include values that are ≥70% (significant values) remove the rest as they are not indicative of strong bootstrap support.

12. What does a tree of all the combined markers look like compared to ITS tree? Perhaps add that to the Suppl information and deposit all alignments into TreeBase. This would make a nice, erudite addition to the paper.

·

Basic reporting

Basic reporting
The text is written in a simple English.
Introduction and background are not very relevant. In the introduction, I miss a hypothesis and/or objectives, questions. These sections are nor really informative enough.
Figures should be labelled on the base of different colours.
Raw data are available from NCBI database

Experimental design

Yes, the manuscript represents an original primary research within Scope of the journal.
The Research question is only mentioned into the abstract.
The investigation uses standard tools commonly applied in DNA sequencing and sequence analyses.
The methods are described with sufficient detail & information to replicate.

Validity of the findings

Impact and novelty are not significant. The authors sequenced some genes, which are largely known to be too conservative.
Data is not robust and the phylogenetic analyses are not really sound. DNA sequences for several species of Lepista available from databases were not included.
Conclusions are in agreement with the evidence. It is difficult to understand the reason of why the authors are not including sequences from other databases in their analyses? Type-sites for some of these species are localized in Europe. In this way, for example, we should find or not depending of the species some intraspecific variation between natural distribution and new distribution site)

Additional comments

The authors should put together sequences from other collections (for example from Europe) available from GenBank and some other database. Some of the species analysed were described originally outside China. For this reason, I’m confident that for developing an accurate barcode approach is imperative to include collections from the complete ranges of geographic distribution of the species analysed, if these have a broad geographic distribution.

Reviewer 3 ·

Basic reporting

The article is well written.

Experimental design

The experimental design is clearly stated.

Validity of the findings

The findings are novel and are suitable to be published in PeerJ.

Additional comments

The manuscript is well written but some small changes should be done before the paper is published. Please carefully check the annotated comments and revise the manuscript accordingly.

Annotated reviews are not available for download in order to protect the identity of reviewers who chose to remain anonymous.

---

## Round 0.2 · accepted · Accept

Thank you for your revision.